# Chemical Modification, Characterization, and Activity Changes of Land Plant Polysaccharides: A Review

**DOI:** 10.3390/polym14194161

**Published:** 2022-10-04

**Authors:** Zhi-Wei Li, Zhu-Mei Du, Ya-Wen Wang, Yu-Xi Feng, Ran Zhang, Xue-Bing Yan

**Affiliations:** College of Animal Science and Technology, Yangzhou University, Yangzhou 225009, China

**Keywords:** plant polysaccharides, chemical modification, preparation, characterization, biological activity

## Abstract

Plant polysaccharides are widely found in nature and have a variety of biological activities, including immunomodulatory, antioxidative, and antitumoral. Due to their low toxicity and easy absorption, they are widely used in the health food and pharmaceutical industries. However, low activity hinders the wide application. Chemical modification is an important method to improve plant polysaccharides’ physical and chemical properties. Through chemical modification, the antioxidant and immunomodulatory abilities of polysaccharides were significantly improved. Some polysaccharides with poor water solubility also significantly improved their water solubility after modification. Chemical modification of plant polysaccharides has become an important research direction. Research on the modification of plant polysaccharides is currently increasing, but a review of the various modification studies is absent. This paper reviews the research progress of chemical modification (sulfation, phosphorylation, acetylation, selenization, and carboxymethylation modification) of land plant polysaccharides (excluding marine plant polysaccharides and fungi plant polysaccharides) during the period of January 2012–June 2022, including the preparation, characterization, and biological activity of modified polysaccharides. This study will provide a basis for the deep application of land plant polysaccharides in food, nutraceuticals, and pharmaceuticals.

## 1. Introduction

Polysaccharides are important biomacromolecules for life’s activities. They are substances composed of different monosaccharides linked by glycosidic bonds. Polysaccharides are classified into three types according to their different sources: plant polysaccharides, animal polysaccharides, and microbial polysaccharides [1]. So far, thousands of polysaccharides have been isolated, among which plant polysaccharides occupy a considerable proportion.

The commonly used method for the extraction of natural polysaccharides includes hot water extraction, ultrasonic-assisted extraction, microwave-assisted extraction, and enzyme extraction [2]. Animal polysaccharides and microbial polysaccharides are usually obtained using enzymatic extraction methods. Plant polysaccharides are mainly obtained by hot-water extraction. At present, the ultrasonic method and microwave assistance have also been widely applied to the three types of polysaccharide extraction to improve the yield of polysaccharides. Microbial polysaccharides and animal polysaccharides are usually surrounded by lipids and must be pre-treated with delipidation before extraction. Some portions of plant tissue have high lipid content and need to be degreased. Afterward, polysaccharide precipitation can be obtained by lowering the dielectric constant of the polysaccharide solution by adding ethanol. Ethanol precipitation is the predominant method for polysaccharide isolation. The methods of ion exchange chromatography, gel filtration, and high-performance liquid chromatography are also commonly used for the separation of polysaccharides. The polysaccharide content of the material varies greatly from different sources, with microbial polysaccharides reaching more than 70%, while plant polysaccharides are generally below 10%.

Plant polysaccharides can be divided into two categories: the first is the substance that stores energy in plants, such as starch, and the second comprises the cell-wall polysaccharides that build plant tissues. Currently studied plant polysaccharides mainly belong to cell-wall polysaccharides. Most plant polysaccharides have antioxidant and immunomodulatory functions [3]. Meanwhile, plant polysaccharides also have various advantages, such as low toxicity [1] and health functions [4], and have been recognized as important food ingredient supplements for healthy food [5]. 

Although natural plant polysaccharides have various biological activities, natural polysaccharides have weak biological activity; some non-water-soluble plant polysaccharides also have poor water solubility [6,7,8], which hinders these polysaccharides’ utilization. Plant polysaccharides are divided into aquatic plant polysaccharides and land plant polysaccharides. Aquatic plant polysaccharides are dominated by algal polysaccharides. Some algal polysaccharides contain sulfate groups [9,10,11]. Researchers have found that the activity of these polysaccharides was significantly higher than land plant polysaccharides [12,13]. Further studies have shown that the attachment of specific foreign functional groups to the polysaccharide structure can change the polysaccharide’s physical properties and biological activity [14]. These findings provide directions for future research on polysaccharides utilization. 

Chemical modification is an important means to modify natural polysaccharides. Through chemical modifications, the weak biological activities of polysaccharides were significantly improved, such as antioxidant and immunomodulatory functions. Some poorly water-soluble polysaccharides also showed significant improvement in their water solubility. Currently, polysaccharide modification methods are classified into five types based on different functional group types (Figure 1), sulfation modification, phosphorylation modification, acetylation modification, carboxymethylation modification, and selenization modification, respectively [15]. Different chemical modification methods lead to different physical and chemical properties obtained by the modified polysaccharides. Some of the modified polysaccharides are not limited to biological effects. These polysaccharides show advantages in preparing new engineering materials by their environmental friendliness and other advantages. The development of environmentally friendly polysaccharide materials will become a hot research topic in the future. 

Recently, natural plant polysaccharide modification has become an important research topic. The intensity of biological activity is influenced by various factors. The polysaccharide source, reagent ratio, reaction time, and reaction temperature can collaboratively affect the final modification performance. Plant polysaccharides’ physical and chemical properties are closely related to foreign groups. However, there is a lack of review articles that summarize the current research progress on plant polysaccharide modifications. It is necessary to summarize the various types of plant polysaccharide modification methods and their corresponding research results in a phased manner. Therefore, this manuscript reviews the modification methods, corresponding characterization methods, and activity changes of natural plant polysaccharides (excluding marine plant polysaccharides and fungi plant polysaccharides) that have been published in recent years (January 2012–June 2022). In total, 51 plant species were included, with plant polysaccharides being extracted from their roots, leaves, fruits, and seeds.

## 2. Polysaccharide Modification

### 2.1. Sulfation of Polysaccharides

Sulfation modification is one of the most prominent methods (Figure 2) [16]. Sulfation modification is the introduction of sulfate groups on the carbon chains of polysaccharides, which usually replace the hydroxyl groups attached to C-2, 4, or 6. Chloro-sulfate pyridine, sulfur trioxide pyridine, and sulfuric acid methods are the three most common methods used for sulfation modification. 

#### 2.1.1. Chloro-Sulfate Pyridine Method

The chloro-sulfate pyridine method is the most-used in sulfation modification. First, chlorosulfonic acid and pyridine are mixed in a certain ratio to form the sulfation reagent. The mixture should be made by slowly dropping chlorosulfonic acid into pyridine and keeping it in an ice bath while mixing. The weighed polysaccharide is dissolved in a specific volume of N-N dimethylformamide or formamide, then slowly added to the sulfation reagent. Afterward, a reasonable reaction time and temperature for the reaction is set (Table 1) [17]. After the reaction, the pH is adjusted to neutral, and the supernatant containing the sulfated polysaccharide is collected. The supernatant is dialyzed and lyophilized to obtain the sulfated polysaccharide solid.

After modification, the sulfated polysaccharide’s physical properties and biological activity are changed. The most important factor affecting the change is the amount of substitution of the sulfate group, called the degree of substitution (DS). The DS is mainly related to the activity of polysaccharide sulfate, and usually, the higher the DS, the stronger the activity [18,19]. The proportion of sulfation reagents, reaction time, and reaction temperature are the main factors that affect the DS. The appropriate test conditions are generally explored by orthogonal or response surface design to obtain the desired DS [20,21].

#### 2.1.2. Sulfur Trioxide Pyridine Method

The test procedure of the sulfur trioxide method is similar to that of the chloro-sulfate pyridine, except that the chloro-sulfate in the sulfation reagent needs to be replaced by the sulfur trioxide compound. The method does not have strict reaction temperature and time requirements (Table 1) [22]. After adjusting the pH to neutral, the supernatant is dialyzed and then freeze-dried to obtain solid sulfated polysaccharide. Compared with the chloro-sulfate pyridine method, sulfur trioxide has a mild reaction process and is easier to obtain highly substituted sulfated polysaccharides, which may make this method appear promising. However, the expensive reagents may limit the broad application of the sulfur trioxide pyridine method.

#### 2.1.3. Sulfuric Acid Method

The sulfuric acid method has less application in polysaccharide modification than the previous two methods. The reaction medium of the concentrated sulfuric acid method has two main types: one is a mixture of concentrated sulfuric acid and pyridine as the reaction medium, and the other is a mixture of concentrated sulfuric acid, n-butanol, and ammonium sulfate as the reaction medium (Table 1) [23]. In the sulfuric acid method, polysaccharide powder can be added directly. The pH is still adjusted to neutral after the reaction. The supernatant is lyophilized by dialysis to obtain solid sulfated polysaccharides. Compared with the chloro-sulfate pyridine method, the reaction conditions of the sulfuric acid method are stable and less affected by time and temperature. However, the strong dehydration effect and strongly acidic environment of sulfuric acid could cause carbonization of polysaccharides and degradation of sugar chains, which affect the yield and modification effect of sulfated polysaccharides [24].

**Table 1 polymers-14-04161-t001:** Sulfated modification of natural plant polysaccharides (2012–2022).

Source	Extraction Methods	B-M_w_ (kda)	A-M_w_ (kda)	Modification Method	Main Modifying Conditions	DS	B-CHO (%)	A-CHO (%)	Biological Activity	Refer
Alfalfa (AP)	Hot-water extraction	22	25	Chloro-sulfonic acid pyridine method	AP (200 mg) N, N-dimethylformamide (20 mL) Ratio of chlorosulfonic acid to pyridine (1.5:1) Reaction at 55 °C for 2.25 h	0.724	90.2	70.3	Antioxidant, Antibacterial Antiobesity	[21]
Opuntia ficus indica cladodes (PC)	Ultrasonic-assisted extraction	7.89	2.1–3.87	Sulfur trioxide pyridine method	PC (400 mg) Formamide (16 mL) Ratio of chlorosulfonic acid to N, N-dimethylformamide (1:6) Reaction at 50 °C for 3 h	0.12–0.46	54.2	21.44–52.72	Anticoagulant	[22]
Amana edulis (AEPS)	Acidic extraction Hot-water extraction	N	N	Sulfuric acid method	AEPS (500 mg) Ratio of concentrated sulfuric acid to n-butanol (3:1) (NH_4_)_2_SO_4_ (125 mg) Reaction at 0 °C for 30 min	1.256–2.134	56.53–65.61	56.48–63.41	Antioxidant	[23]
Persimmon fruits (PFP)	Hot-water extraction	130	48–53	Chlorosulfonic acid pyridine method	PAS (500 mg) Formamide (20 mL) Ratio of chlorosulfonic acid to pyridine (1:8, 1:4, 1:2) Reaction at 50 °C for 2 h	0.8–2.5	N	N	Immunomodulatory	[25]
Longan (LP)	Hot-water extraction	118	105	Sulfuric acid method	LP (500 mg) Ratio of sulfuric acid to butanol complex (3:1) Ammonium sulfate (125 mg) Reaction at 10 °C for 3 h	2.011	N	N	Immunomodulatory Antitumor	[26]
Dendrobium huoshanense (DHPD)	Hot-water extraction	8.09 × 10^3^	1.01–1.10 × 10^4^	Chlorosulfonic acid pyridine method	DHPD (500 mg) Formamide (5 mL) Ratio of chlorosulfonic acid to pyridine (1:2) Reaction at 60 °C for 30 min, 60 min	0.475–0.94	92.89	37.7–56.35	Antiglycation	[27]
Astragalus (APS)	N	N	N	Chlorosulfonic acid pyridine method	APS (400 mg) Ratio of chlorosulfonic acid to pyridine (1:6) Reaction at 95 °C for 1 h	1.4	97	N	Immunomodulatory	[28]
Cyclina sinensis (CSPS-1)	Hot-water extraction	N	N	Chlorosulfonic acid pyridine method	ASP (100 mg) N, N-dimethylformamide (20 mL) Ratio of chlorosulfonic acid to pyridine (1:4) Reaction at 65 °C for 2 h	0.3–1.02	N	N	Antitumor	[29]
Lyciumbarbarum (LBPS)	N	11.87–13.12	18.58–29.06	Chlorosulfonic acid pyridine method	LBPS (400 mg) Ratio of chlorosulfonic acid to pyridine (1:8) Reaction at 80 °C for 2 h	N	N	35.37–63.21	Immunomodulatory	[30]
Artemisia sphaerocephala (ASP)	Microwave-assisted extraction	73.48	18.06–32.72	Chlorosulfonic acid pyridine method	ASP (500 mg) Formamide (20 mL) Ratio of chlorosulfonic acid to pyridine (2:1) Reaction at 60 °C for 3 h	0.44–0.63	90.2	N	Antitumor	[31,32]
Cyclocarya paliurus (CP)	Hot-water extraction	1.16 × 10^3^	0.97–1.07 × 10^3^	Chlorosulfonic acid pyridine method	CP (600 mg) Formamide (60 mL) Ratio of chlorosulfonic acid to pyridine (1:4, 1:8) Reaction at 60 °C for 4 h	0.12–0.42	N	42.41–63.77	Antioxidant Immunomodulatory	[33]
Sphallerocarpus gracilis (SGP)	Hot-water extraction	218	59	Chlorosulfonic acid pyridine method	SGP (200 mg) Formamide (15 mL) Ratio of chlorosulfonic acid to pyridine (1.3:1) Reaction at 65 °C for 3.4 h	0.99	N	N	Antioxidant	[34]
Borojoa sorbilis cuter (BP)	Ultrahigh pressure extraction	35.8	N	Sulfur trioxide pyridine method	BP (100 mg) DMSO (5 mL) SO_3_⋅Pyr (400 mg) Reaction at 60 °C for 7 h	1.18	N	N	Antitumor	[35]
Cyclocarya paliurus (CP)	Hot-water extraction	1.16 × 10^3^	0.97–1.12 × 10^3^	Chlorosulfonic acid pyridine method	CP (600 mg) CSA to Pyr at Ratios of 1:1, 1:4, 1:6, and 1:8 N, N-dimethylformamide (60 mL) Reaction at 60 °C for 4 h	0.12–0.55	60.62	35.87–49.71	Antioxidant	[36]
Cyclocarya paliurus (CP)	Hot-water extraction (pretreatment degreasing)	139	212	Chlorosulfonic acid pyridine method	CP (600 mg) Formamide (20 mL) Ratio of chlorosulfonic acid to pyridine (1:4) Reaction at 60 °C for 4 h	0.42	63.77	42.41	Anti-inflammatory Antioxidant	[37]
Blackcurrant (BCP)	Ultrasonic-assisted extraction	17.6 × 10^3^	17.6–18.5 × 10^3^	Aminosul-fonic acid method	BCP (100 mg) N, N-dimethylformamide (40 mL) 4-dimethylamino-pyridine (162.2 mg) Ratio of amino sulfonic acid to BCP (8:1 to 30:1) reaction at different temperatures (60–100 °C) for various periods (1–5 h)	0.53–1.28	84.89	N	Antioxidant Hypoglycemic activity	[38]
Artemisia sphaerocephala (PAS)	Microwave-assisted extraction	139.8	103–760	Chloro-sulfonic acid pyridine method	PAS (500 mg) Formamide (30 mL) Ratio of chlorosulfonic acid to pyridine (2:1) Reaction at 60 °C for 15 to 300 min	0.63–0.86	N	N	Antitumor	[39]
Pumpkin (N)	Hot-water extraction	10.18	3.84–7.7	Chlorosulfonic acid pyridine method	Polysaccharide (250 mg) Formamide (10 mL) Reaction at 60 °C for 3 h	0.26–0.45	95.17	7.53–12.29	Anticoagulant	[40]
Abelmoschus manihot (Linn.) Medicus (LAMP)	Hot-water extraction	N	264.2–1044.2	Aminosul-fonic acid method	Polysaccharide (40 mg) N, N-dimethylformamide (15 mL) Aminosul-fonic acid (40, 80, 120 mg) Reaction at 80 °C for 3 h	0.25	99.76	52.1	Immunomodulatory	[41]
Bupleurum chinense (BCP)	Hot-water extraction	29	37.6–51.7	Chlorosulfonic acid pyridine method	BP (200 mg) Ratio of chlorosulfonic acid to pyridine (1:4, 1:8) Reaction at 80 °C for 2 h	0.38–0.61	97.5	N	Antioxidant Antisenescence	[42]
Polygonatum sibiricum (N)	Hot-water extraction	132.6	82.1–117.0	Sulfur trioxide pyridine method	Reaction at 80 °C for 1 h, 3 h, 6 h	0.5–1.9	94	78.5–88	Immunomodulatory	[43]
Lycium barbarum L. (LBP)	Enzyme extraction	80.00	131.78	Chlorosulfonic acid pyridine method	LBP (100 mg) Formamide (5 mL) Ratio of chlorosulfonic acid to pyridine (3:1) Reaction at 60 °C for 3 h	1.43	N	N	Antiangiogenic	[44]
Cucumber (N)	Hot-water extraction	N	N	Chloro-sulfonic acid pyridine method	Polysaccharide (500 mg) N, N-dimethyl formamide (10 mL) Reaction at 80 °C for 3 h	0.65	20.6	31.2	Antioxidant	[45]
Pumpkin (N)	Hot-water extraction	N	N	Chloro-sulfonic acid pyridine method	polysaccharide (500 mg) N, N-dimethylformamide (30 mL) Ratio of chlorosulfonic acid to pyridine (2:5) Reaction at 100 °C for 1 h	0.35	81	61.8	Antioxidant	[46]
Chinese yam (CYP)	Enzyme-assisted hot-water extraction	19.5	29.6	Chloro-sulfonic acid pyridine method	Ratio of chlorosulfonic acid to pyridine (1:5) Reaction at 70 °C	0.44	35.77	33.27	Immunomodulatory	[47]
Jerusalem artichoke (JAP)	N	2.6	N	Sulfur trioxide pyridine method	JAP (200 mg) Pyridine (5 mL) SO_3_⋅Pyr (600 mg) Reaction at 95 °C for 4 h	0.56	N	N	Antitumor	[48]
Chinese yam (CYP)	Enzymatic-assisted extraction	33.33	37.04	Chloro-sulfonic acid pyridine method	CYP (400 mg) Formamide (100 mL) Ratio of chlorosulfonic acid to pyridine (1:3) Reaction at 70 °C for 3 h	0.51	47.45	36.77	Immunomodulatory	[49]
Cyclocarya paliurus (CPP, CPP_0.05_)	Hot-water extraction (pretreatment degreasing)	35.8 30.1	12.64–52.62	Chloro-sulfonic acid pyridine method	CPP, CPP_0.05_ (20 mg) Formamide (20 mL) Ratio of chlorosulfonic acid to pyridine (1:6) Reaction at 60 °C for 2 h	0.18–0.32	62.75	50.14–54.42	Immunomodulatory	[50]
Tamarind seed (TSP)	Hot-water extraction	1370	1340	Sulfur trioxide pyridine method	TSP (1 g) Reaction at 50 °C for 4 h	0.31	N	N	Osteogenic activities	[51]
Jujube (JP)	Hot-water extraction	275	317	Chloro-sulfonic acid pyridine method	JP (500 mg) Formamide (100 mL) Ratio of chlorosulfonic acid to pyridine (1:1) Reaction at 75 °C for 1 h	0.664	75.4	63.40	Antioxidant Antibacterial	[52]
Orchis chusua D. Don (SP)	Hot-water extraction	369	318	Sulfur trioxide pyridine method	SP (200 mg) N, N-dimethylformamide (20 mL) SO_3_⋅Pyr (100 mg) Reaction at 80 °C for 3 h	0.12	47.93	71.55	Antioxidant Probiotic ability	[53]
Cyclocarya paliurus (CP)	Hot-water extraction	139.6	161.5	Chloro-sulfonic acid pyridine method	CP sample (400 mg) formamide (40 mL) Ratio of chlorosulfonic acid to pyridine (1:7) Reaction at 60 °C for 4 h	0.17	N	N	Antioxidant Antitumor	[54]
Cardamine hupingshanensis (CHP)	Hot-water extraction	N	22.2	Sulfur trioxide pyridine method	CHP (600 mg) DMSO (180 mL) SO_3_⋅Pyr (15 g) Reaction at 55 °C for 2 h	N	N	N	Antioxidant	[55]

N: not referred; B-M_w_: M_w_ before modification; A-M_w_: M_w_ after modification; B-CHO: carbohydrate content before modification; A-CHO: carbohydrate content after modification.

### 2.2. Selenization of Polysaccharides 

Selenium is an essential element for human life activities, and a lack of selenium in the body can produce various diseases. Proper selenium supplementation can enhance antioxidant capacity and improve immunity [56]. The combination of selenium and polysaccharide forms a selenized polysaccharide with the dual activity of selenium and polysaccharide, which is more easily absorbed and utilized by the human body [57]. Selenization of polysaccharides involves the addition of inorganic selenium to the polysaccharide carbon chain under acidic conditions to produce absorbable organic selenium (Figure 3) [58]. The common utilization methods of selenization are the glacial acetic acid/sodium selenite, nitric acid/sodium selenite, and selenium oxychloride methods.

#### 2.2.1. Glacial Acetic Acid/Sodium Selenite Method 

The glacial acetic acid/sodium selenite is a relatively simple method for selenization modification. The polysaccharide is dissolved in glacial acetic acid at room temperature with constant stirring until completely dissolved. Then, a suitable amount of sodium selenite is added, and the selenized polysaccharide solution is obtained after the heating reaction (Table 2) [59]. The selenized polysaccharide is precipitated with ethanol after cooling, and the selenized polysaccharide is obtained by dialysis and freeze-drying of the solution after the precipitate was redissolved in water.

#### 2.2.2. Nitric Acid Sodium/Selenite Method 

Nitric acid sodium/selenite is the most common method used for selenization modification. The polysaccharide is added to the HNO_3_ solution and stirred until completely dissolved, and then BaCl_2_ is added. The Na_2_Se_2_O_3_ is added to the polysaccharide nitric acid solution with stirring to control the reaction temperature [60]. After the reaction, the mixture is cooled to room temperature. The pH is adjusted with a saturated sodium carbonate solution. Then, sodium sulfate is added to remove Ba^2+^, and the supernatant is retained by centrifugation. After testing the supernatant with ascorbic acid for the absence of Na_2_Se_2_O_3_, the selenized polysaccharide is obtained after freeze-drying by dialysis.

#### 2.2.3. Selenium Oxychloride Method

The selenium oxychloride method is commonly used for the selenization of polysaccharides with low water solubility. The polysaccharide is dissolved in pyridine, and selenium chloride is added for the reaction (Table 2) [61]. After the reaction, the polysaccharide is precipitated with ethanol, and the precipitate is collected by centrifugation and redissolution. The redissolved solution is dialyzed and lyophilized to obtain selenized polysaccharides. The reagents used in the selenium chloride method are relatively expensive and difficult to promote, so this method is usually used only in the laboratory.

### 2.3. Phosphorylation of Polysaccharides 

Natural phosphorylated plant polysaccharides rarely appear. Therefore, phosphorylation modification (Figure 4) is an important approach for obtaining phosphorylated polysaccharides [16]. 

Phosphorylated phosphate, phosphate, and phosphate/anhydride are three commonly used phosphorylation modification methods.

#### 2.3.1. Phosphate Method 

The phosphorylation reagent is made by mixing sodium tripolyphosphate and trimetaphosphate in a specific ratio. After that, the pH of the phosphorylation reagent is adjusted to 7–9. Polysaccharides and sodium sulfate are added to the phosphorylation reagent and reacted at a specific temperature to produce phosphorylated polysaccharides (Table 3) [83]. The supernatant containing phosphorylated polysaccharides is lyophilized by dialysis to obtain the phosphorylated polysaccharides. 

#### 2.3.2. Phosphoric Acid Method

Urea is melted at a high temperature to make a urea solution. Afterward, polysaccharides and phosphoric acid are added to the urea solution [84]. After a high-temperature reaction, the phosphorylated polysaccharide is produced, and the supernatant is collected after centrifugation (Table 3). The phosphorylated polysaccharide is obtained by dialysis and lyophilization. 

#### 2.3.3. Phosphorus Oxychloride Method 

The phosphorus trichloride method can prepare the phosphorylation of polysaccharides with high DS. The polysaccharide is dissolved with N, N-dimethylformamide, and POCl_3_ and pyridine are mixed in a specific ratio to make the phosphorylation reagent (Table 3) [85]. The phosphorylation reagent is slowly added to the polysaccharide solution, and the reaction is carried out at a set temperature and time. As soon as the reaction is complete and has cooled to room temperature, the supernatant is centrifuged, and the polysaccharide is precipitated with ethanol. The precipitate is redissolved in water, dialyzed, and then lyophilized to obtain phosphorylated polysaccharides.

### 2.4. Carboxymethylation of Polysaccharides 

Carboxymethylation modification is a modification method that introduces carboxymethyl into the sugar chain (Figure 5) [92]. The sodium hydroxide reacts with the hydroxyl group of the polysaccharide to form an alcohol oxygen group, which is followed by addition of monochloroacetic acid to form carboxymethyl. Compared with other modification methods, the carboxymethylation method is simple and low-cost and is one of the most widely utilized polysaccharide modification methods. Depending on if the polysaccharide is dissolved in organic reagents in advance or not, the carboxymethylation modification methods are classified into water and solvent methods.

#### 2.4.1. Water Media Method 

The aqueous medium method involves dissolving polysaccharide powder directly with sodium hydroxide solution, followed by adding monochloroacetic acid to react at a specific temperature to synthesize carboxymethyl polysaccharide (Table 4) [53]. The water medium method has low reagent utilization, a high chance of side reactions, and difficult post-treatment, resulting in the aqueous medium method being used much less frequently than the solvent method.

#### 2.4.2. Solvent Method 

The solvent method is to dissolve the polysaccharide powder with the organic solvent, followed by the slow addition of sodium hydroxide solution (Table 4) [93]. The subsequent steps are the same as the water medium method. The solvent method has the advantages of a uniform reaction and a short reaction time. Thus, this method is the most widely used method for carboxymethylation modification.

### 2.5. Acetylation of Polysaccharides

The acetylation modification is less-used in polysaccharide modification, but it is also a modification method that cannot be ignored (Figure 6) [15]. The most important method of acetylation modification is the acetic anhydride method.

#### Acetic Anhydride Method 

The polysaccharide is dissolved in an organic solvent and reacted by adding an acetylation reagent (acetic acid or acetic anhydride). The pH of the reaction is kept above 7.0, and acetylated polysaccharides are formed after the reaction at a specific temperature (Table 5) [98]. Acetylated polysaccharides can be obtained by ethanol precipitation or direct dialysis lyophilization.

## 3. Characterization of Polysaccharides

The structure of the modified polysaccharide needs to be identified after polysaccharide modification. The degree of substitution (DS), molecular weight (M_w_), Fourier infrared spectroscopy (FT-IR spectra), nuclear magnetic resonance (NMR), and monosaccharide composition analysis are all important tools needed to characterize the polysaccharide structure. The results of these characterizations can be used to obtain important information, such as the DS of foreign groups, whether the modification was successful, and the position of the active group attached to the sugar chain.

### 3.1. DS 

The DS is the number of substituents per unit of monosaccharide in the polysaccharide. The biological activity of modified polysaccharides is usually positively correlated with DS. In most cases, the optimal polysaccharide modification conditions can be obtained using orthogonal design and response surface design to obtain the desired highly substituted polysaccharide.

The DS of phosphorylated polysaccharides is determined using the molybdenum blue colorimetric method, and the DS was calculated [105] according to the following Equation (1):DS = (162 × P)/(3100 − 84 × P)(1)
where P is phosphate radical content.

The DS of sulfated polysaccharides is determined by hydrolyzing the polysaccharides under acidic conditions and determining the sulfur content of the samples using the barium sulfate turbidimetric method. The DS is calculated [106] according to the following Equation (2):DS = (1.62 × S)/(32 − 1.02 × S)(2)
where S is phosphate radical content.

The DS of carboxymethylated polysaccharides is determined by neutralization titration. The DS is calculated [92] according to the following Equations (3) and (4):A = (V_0_C_0_ − V_1_C_1_)/W(3)
DS = 0.162A/(1 − 0.058A)(4)
A: Consumption of sodium hydroxide per gram of sampleV_0_: Consumption of sodium hydroxideC_0_: Concentration of sodium hydroxideV_1_: Consumption of hydrochloric acidC_1_: Concentration of hydrochloric acidW: Weight of sample

The DS of acetylated polysaccharides is determined by neutralization titration. The DS is calculated as follows (5) and (6) [107]:DS = 162 × AC/4300 − 42 × AC(5)
AC = [(V_1_ − V_2_) × M × 0.043] × 100/W(6)
V_1_: Volume of hydrochloric acid required for the polysaccharide before modificationV_2_: Volume of hydrochloric acid required for the polysaccharide sample after modificationM: Molar concentration of hydrochloric acidW: Weight of sample

Unlike the remaining four modified polysaccharides, the selenized polysaccharides directly determine the selenium content in polysaccharides using inductively coupled plasma atomic emission spectrophotometry or atomic fluorescence spectrometry [72,73].

### 3.2. M_w_

The M_w_ of plant polysaccharides ranges from a few thousand to several million. After chemical modification, the M_w_ of polysaccharides changes significantly. M_w_ is an important factor affecting polysaccharide activity [103]. Typically, polysaccharide modification occurs under acidic conditions, which causes degradation of the polysaccharide, resulting in a decrease in M_w_. However, the M_w_ of the introduced substituent is usually much higher than that of the hydroxyl group in the substituent position, increasing the M_w_ of the modified polysaccharide. Therefore, further studies on the correlation between M_w_ and biological activity are needed. The determination of M_w_ of polysaccharides is mainly performed by high-performance liquid chromatography (HPLC) or high-performance gel permeation chromatography (HPGPC). These two methods have the advantages of simplicity, rapidity, and good reproducibility and have been widely used to determine the molecular weight of macromolecular substances.

### 3.3. FT-IR Spectra 

FT-IR spectra are compared before and after modification to confirm whether or not the polysaccharide is successfully modified. The FT-IR spectra of the modified polysaccharides show new characteristic peaks after introducing new groups. A sulfated polysaccharide shows a characteristic peak near 1250 cm^−1^, caused by S=O.Another characteristic peak will appear near 810 cm^−1^ due to C-O-S symmetry vibration [48,54]. After the polysaccharide is attached to the phosphate group, characteristic peaks appear near 1033 and 898 cm^−1^, attributed to the P-OH and P-O-C bonds, respectively. The third peak appears near 1245 cm^−1^ due to the asymmetric stretching vibration of the P=O bond [105]. Carboxymethylated polysaccharides show characteristic peaks near 1600 cm^−1^, 1420 cm^−1^, and 1330 cm^−1^, indicating the presence of -COO groups [92]. Acetylated polysaccharides show characteristic peaks near 1730 cm^−1^ and 1240 cm^−1^, which are caused by C = O and C-O-C [104,107]. The selenated polysaccharides exhibit characteristic peaks at 930 cm^−1^ and 830 cm^−1^ due to C-O-Se stretching vibration and Se=O asymmetric stretching [61,74].

### 3.4. Monosaccharide Composition

Polysaccharides are composed of different monosaccharides linked by glycosidic bonds, and different monosaccharide ratios affect the biological activity of polysaccharides. Capillary electrophoresis (CE), high-performance liquid chromatography (HPLC), and high-performance anion exchange chromatography (HPAEC) are the three commonly used methods for analyzing the composition of monosaccharides.

CE has the advantages of rapid separation, high sensitivity, and requiring only a small amount of sample and is mainly used for the analysis and detection of monosaccharides and oligosaccharides [108].

HPLC has high efficiency, speed, sensitivity, and selectivity. There are various types of detectors for HPLC, and the main ones used for monosaccharide detection are universal and UV–visible absorption detectors [109]. Universal detectors include evaporative light scattering and differential detectors, which can directly detect monosaccharide molecules. However, general-purpose detectors respond not only to sugars but also to amino acids, lipids, carbohydrates, and other substances. UV–visible absorption is the more widely used detection method for the high-performance liquid phase. Monosaccharides have no UV absorption and require derivatization (pre-column derivatization or post-column derivatization) before they can be recognized by the UV detector. High-performance liquid chromatography with pre-column derivatization is a common method for analyzing polysaccharides.

The HPAEC method allows the direct detection of polysaccharides without derivatization. The assay is sensitive, fast, and accurate. Polysaccharides contain polyhydroxy groups, which are electrochemically active and have a dissociative capacity [110]. High pH drench solution (sodium hydroxide solution) exists partially or completely as anions and can be retained and separated on an anion exchange column. Different anions have different abilities to compete with OH^−^ for the binding sites of the stationary phase, leading to different retention times and thus to separation. The retention of sugar compounds on anion-exchange separation columns depends mainly on the number of charges, molecular size, composition, and structure. 

### 3.5. NMR

Nuclear magnetic resonance (NMR) is widely used for the structure elucidation of polysaccharides and their derivatives. The ^13^C NMR is a useful tool for tracking polysaccharide substitution reactions. The result of polysaccharides modification also can be confirmed by the NMR spectroscopic analysis. The sulfate group usually undergoes non-selective substitution at C-2,4 or 6 [106], and new peaks may be observed around 68–80 ppm and 100–110 ppm [22]. The peak of the phosphate group usually occurs around 60–90 ppm. These findings indicate phosphorylation occurs at C-2, 3, 6 [105]. In the ^13^C NMR spectrum, characteristic peaks of carboxymethyl carbonyl and methylene groups can be observed in the region of 176–180 and 70–71 ppm, respectively [92]. Selenization of polysaccharides mainly occurrs in C-6, and new peaks are observed around 60–65 ppm [67,68]. The acetylated polysaccharide has two absorption peaks near 175 ppm (carbonyl of O-acetyl group) and 21.5 ppm (methyl of O-acetyl group), which can prove the success of the acetylation modification [101].

## 4. Biological Activity

Natural plant polysaccharides have anti-inflammatory, hypoglycemic, antioxidant, and immunomodulatory functions. However, most plant polysaccharides possess only a few functions and low activity. Plant polysaccharides that have been chemically modified show significant improvements in all functions. Some plant polysaccharides were modified to obtain new biological activities.

### 4.1. Bioactivity of Sulfated Polysaccharides

Sulfation modification is the most frequently applied method in polysaccharide modification. This may be since sulfated polysaccharides possess a variety of biological functions, including antitumoral, antioxidative, immunomodulatory, anticoagulative, and anti-aging (Table 1). The sulfate group is hydrophilic, so the water solubility of the sulfated polysaccharide is also significantly improved. 

Tumors are a serious medical problem worldwide. Tumors are divided into benign and malignant tumors, and malignant tumors can develop into cancer and directly threaten human life. Tumor angiogenesis is an important form of tumor expansion, and inhibition of tumor angiogenesis is an important tool needed in tumor treatment [111]. Studies have shown that sulfated polysaccharides inhibit tumor growth through two main pathways. The first pathway inhibits tumor growth by inhibiting vascular endothelial cell proliferation and reducing vascular endothelial cell migration with tumor blood vessel formation [44,112]. The second approach is to interfere with the normal physiological activities of tumor cells at specific stages of the cell cycle [39,54]. Among these studies, the M_w_ and polydispersity of sulfated polysaccharides were important factors affecting the antitumor effect. 

The immune system consists of immune organs, immune cells, and active immune factors, which are important for the body to resist the invasion of foreign antigens and remove harmful substances from the body. Polysaccharides enhance immune function mainly by activating immune organs and immune cells. After sulfation modification, the immunomodulatory ability of polysaccharides was significantly enhanced. The CPP and S-CPP enhanced the immunity through MAPK and NF-κB signaling pathway by triggering TLR2/4 [50]. The immunomodulatory function of sulfated modified polysaccharides was significantly stronger than that of unmodified polysaccharides. On the other hand, the S-CYP promotes the proliferation of splenic lymphocytes and induces the differentiation of splenic lymphocytes into T-lymphocytes in synergy with ConA [49]. The sulfate group plays an important role in the immune-enhancing process. The sulfate group facilitates polysaccharide binding to immune cell surface receptors through electrostatic interactions and hydrogen bonding, thus activating immune cells and enhancing immunity [26]. 

Natural plant polysaccharides have weak or no anticoagulant effect. After sulfated modification, sulfated polysaccharides exhibit anticoagulant activity by prolonging activated partial thromboplastin time (APTT) and thromboplastin time (TT), which should be attributed to the sulfate groups [23,40]. Related experiments have also shown that the anticoagulant activity of sulfated polysaccharides is determined by the content of the sulfate group in the polysaccharide [113,114]. The antioxidant capacity of sulfated polysaccharides was significantly stronger than that of unmodified polysaccharides. Antioxidant capacity is positively correlated with the hydroxyl groups on the sugar chains [115], which are partially inaccessible due to the spatial structure of polysaccharides. After sulfation modification, the hydroxyl groups inside the polysaccharide are released, which enhances the scavenging ability of the polysaccharide against the superoxide anion. The content of sulfate groups is an important factor affecting the antioxidant capacity of sulfated polysaccharides [34]. This may be because sulfate groups’ electrons can interrupt the free radicals’ chain reaction [38]. Several studies confirm that sulfated polysaccharides can significantly down-regulate the p53-p21 and p16-pRb levels to slow oxidative stress-induced cellular senescence [42].

### 4.2. Bioactivity of Selenized Polysaccharides

Selenization-modified plant polysaccharides possess both selenium and polysaccharide bioactivities. Selenated polysaccharides possess antioxidant, immunomodulatory, antitumor, and hypoglycemic functions (Table 2). 

Se-RAPS-2 showed superior antioxidant activity to RAPS-1 and RAPS-2 in HepG2 cells, and the scavenging effect of Se-RAPS-2 on DPPH and ABTS radicals was superior to that of RAPS-1 [73]. This is consistent with previous findings that selenized polysaccharides have higher antioxidant activity [116,117]. The hydrogen supply capacity is an important factor in assessing the antioxidant capacity of polysaccharides. The hydrogen atom of heteropolymeric carbon in Se-RAPS-2 is activated by the selenite group, and Se-RAPS-2 obtains a stronger hydrogen supply capacity. This explains why Se-RAPS-2 exhibits better antioxidant activity than RAPS-2. The Se-MCPIIa-1 significantly reduced fasting blood glucose levels and increased insulin levels and antioxidant enzyme activity in diabetic mice [46]. The sCPPS can synergize with PHA or LPS to promote lymphocyte proliferation and increase the ratio of CD^4+^ to CD^8+^ in T cells [75]. Compared to CPPS, sCPPS significantly increased serum levels of immune factors. This implies that the selenization modification enhanced the immunomodulatory activity of CPPS. The high selenium content of SePAS upregulated the phosphorylation levels of ERK, JNK, and p38. It promoted the proliferation and phagocytosis and increased the levels of IL-6, NO, TNF-α, and IL-1β in RAW264.7 cells [75]. Wang [70] showed that selenium content is a key factor affecting the in vitro antitumor activity of selenated polysaccharides, which predicted that increasing the selenium content in polysaccharides will be an important future research direction. 

### 4.3. Bioactivity of Phosphorylated Polysaccharides

Phosphorylation modification enhances polysaccharides’ antioxidant, antiviral, and immunomodulatory abilities (Table 3). The phosphate group can activate the hydrogen atom of the heteropolymeric carbon [85]. Thus, phosphorylated polysaccharide becomes a stronger hydrogen atom donor and can provide activated hydrogen atoms for scavenging free radicals. After phosphorylation modification, P-CP exhibited significantly stronger antioxidant capacity than CP [83]. P-CP significantly increased the content of the SOD and decreased the content of MDA in oxidatively damaged cells. In terms of stimulating the production of ROS and antioxidant enzymes, inhibiting lipid peroxidation, and reducing apoptosis capacity, P-CP also showed more effectiveness than CP. PAEPs and PDIP also show similar antioxidant capacities [23,84]. 

The antiviral mechanism of polysaccharides is that the polysaccharide occupies a position on the viral membrane, forming a stable polysaccharide complex that prevents viral adsorption. Feng found that the antiviral ability of PRCP with higher phosphorylation levels was significantly enhanced, independent of the glyoxylate content. It showed that the phosphorylation level is one of the key factors affecting the antiviral ability of phosphorylated polysaccharides. On the other hand, Ming found that PCIPS was more effective than CIPS in suppressing viral gene replication [87]. PRCPS has shown good immunomodulatory effects as a vaccine adjuvant or direct application [88]. As an adjuvant for the FMD vaccine, pRCPS promoted the killing activity of cytotoxic T lymphocytes and natural killer cells. Importantly, pRCPS enhanced the expression of MHCII, CD^40+^, CD^86+^, and CD^80+^ in dendritic cells [88]. In treating cyclophosphamide-induced immunosuppressed mice, PRCP significantly increased serum immunoglobulin concentrations and serum cellular immune factor levels. Meanwhile, PRCP increased the ratio of selected T-cell subpopulations (CD^3+^, CD^4+^, and CD^4+^ to CD^8+^ ratios) [90]. The phosphorylation of polysaccharides promotes their immune-enhancing effects. 

### 4.4. Bioactivity of Carboxymethylated Polysaccharides

Carboxymethylation modification enhanced polysaccharides’ antioxidant, hypoglycemic, immunomodulatory, and antibacterial abilities, and some of them could inhibit α-amylase activity after modification (Table 4). 

Compared to natural polysaccharides, carboxymethylated polysaccharides possess good water solubility, which is attributed to introducing a hydrophilic group (carboxymethyl group). The carboxymethylation modification introduces negatively charged carboxyl groups to build negatively charged hydrophilic surface structures for polysaccharides. It greatly improves the water solubility of carboxymethylated polysaccharides and helps to enhance antioxidant activity [118]. Compared with CP, CM-CP effectively protected RAW264.7 from H_2_O_2_-induced damage [54]. The CM-CP reduced the secretion of lactate dehydrogenase, reactive oxygen species, and malondialdehyde, increasing superoxide dismutase levels. CM-CP simultaneously inhibited the abnormal apoptosis of cells. Some have argued that the anti-apoptotic effect of CM-CP is closely associated with down-regulation of Caspase-9/3 activity and alleviation of S-phase cell cycle arrest. Duan found that CRNPs have good hydroxyl radical, superoxide radical scavenging ability, and better anti-lipid peroxidation activity [93]. Moreover, it was shown that the activity of CRNPs increased significantly with DS. Zhao [97] showed that the carboxymethylated modified polysaccharides not only acquired better antioxidant capacity but also exhibited better immunomodulatory activity and reduced polychlorinated biphenyl-126-induced shrinkage of immune organs in mice. Carboxymethylated polysaccharides inhibit bacteria by inducing damage to the bacterial cell wall and cytoplasmic membrane [92]. Song [119] confirmed this conclusion by analyzing the inhibitory effect of polysaccharides in carboxymethylated barley bran on Staphylococcus aureus. The corn silk polysaccharides were subjected to carboxymethylation modification, sulfation modification, and acetylation modification, respectively. The results showed that the carboxymethylated corn silk polysaccharide had higher α-amylase inhibitory activity than the remaining polysaccharides. Therefore, carboxymethylated polysaccharides have the potential to become hypoglycemic agents. 

### 4.5. Bioactivity of Acetylated Polysaccharides

Acetylated polysaccharides mainly possess antioxidant activity and immunomodulatory functions, and the emulsification of acetyl polysaccharides has been significantly improved. (Table 5). The introduction of acetyl groups in polysaccharides weakens the dissociation energy of O-H bonds and therefore has a greater ability to provide hydrogen [99]. The CPP_0.1_ and Ac-CPP_0.1_ significantly increased the levels of superoxide dismutase, glutathione peroxidase, and catalase on hydrogen peroxide-treated dendritic cells [101]. Meanwhile, both CPP_0.1_ and Ac-CPP_0.1_ up-regulated the expression of Nrf2 and down-regulated the Keap1. However, Ac-CPP_0.1_ had a better effect on antioxidant capacity than CPP_0.1_. Acetylation modification also significantly enhanced the inhibitory effect of polysaccharides on lipid peroxidation [102]. Liu found that acetylated polysaccharides significantly stimulated macrophage proliferation and enhanced macrophage activation [100]. This demonstrated the good immunomodulatory function of acetylated polysaccharides. It also indicates the potential application of acetylated polysaccharides as an adjuvant for immunotherapy. Multiple studies found that acetylation modifications significantly improved the emulsification of polysaccharides [98,106,107]. This contributes to a wider range of application categories.

## 5. Conclusions

Based on the studies in this review, all five modification methods improved the antioxidant and immunomodulatory effects of plant polysaccharides. Of course, each of the different modification methods has its characteristics. For example, sulfation mainly improved the antioxidant and immunomodulatory properties of polysaccharides, carboxymethylation mainly improved the water solubility of polysaccharides, and acetylation mainly enhanced the emulsification of polysaccharides. Therefore, the corresponding modification methods can be chosen for different needs. However, the structure of polysaccharides is complex, and chemical modification may negatively affect some polysaccharides [96], and plant polysaccharides should not be modified blindly. 

Meanwhile, there are reports about multiple modifications of plant polysaccharides [102,104], in which two or more modifications of one polysaccharide are performed. There are few relevant studies, and further research is still needed to determine whether multiple modifications can achieve a superimposed effect. The solventless method is an emerging method for polysaccharide modification, which is environmentally friendly, non-toxic to humans, and has broad application prospects. The modified polysaccharides prepared by the solventless method have shown great advantages in preparing new materials and pharmaceuticals [120,121]. Under the current green theme, the solventless method will be applied to polysaccharide modification more frequently.

The application of modified plant polysaccharides is still at the cellular and mouse model stage. Therefore, future research on modified plant polysaccharides should be advanced in the clinical direction. This includes purification of modified plant polysaccharides, safety assessment, production of polysaccharide products, and in vivo release of polysaccharide activity. Polysaccharides have been recognized as a health food. The application of modified polysaccharides in food is still relatively rare. The antioxidant, hypolipidemic, immunomodulatory ability, and good emulsifying properties of modified polysaccharides suggest that modified polysaccharides can have a wide range of applications in the food field. 

In recent years, natural green products have been gaining attention. As an important natural green product, the opportunities that plant polysaccharides offer should be seized upon to draw more attention to their advantages. Chemical modification, as an important tool to improve plant polysaccharides’ properties, requires further research.

## Figures and Tables

**Figure 1 polymers-14-04161-f001:**
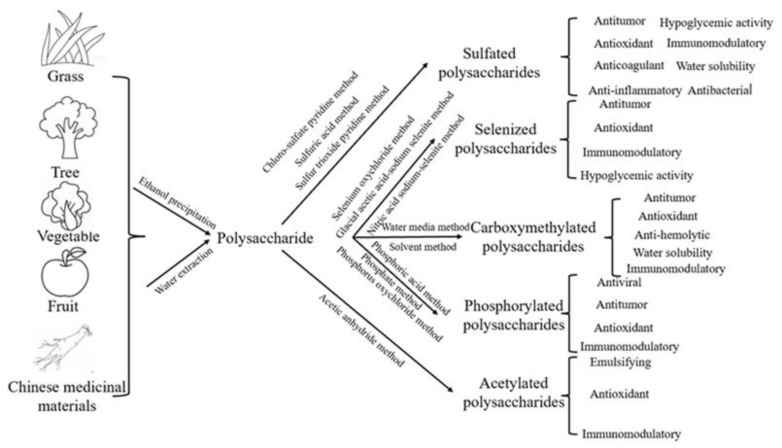
Polysaccharide modification.

**Figure 2 polymers-14-04161-f002:**
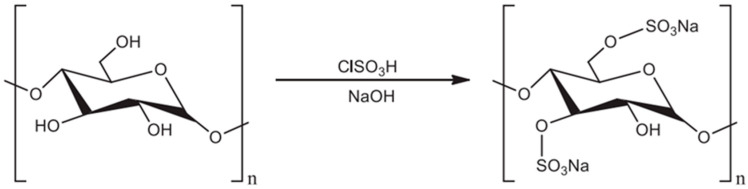
Sulfation of polysaccharides.

**Figure 3 polymers-14-04161-f003:**
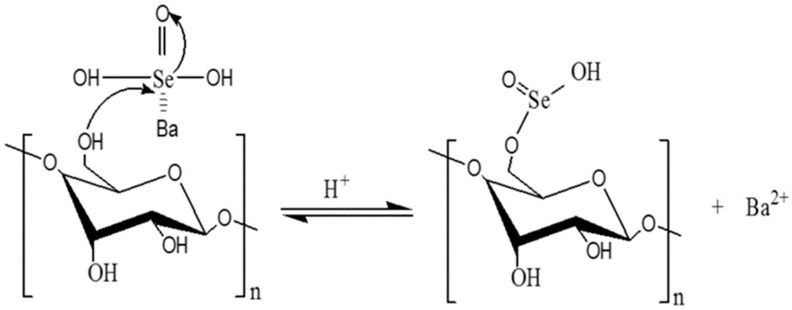
Selenization of polysaccharides.

**Figure 4 polymers-14-04161-f004:**
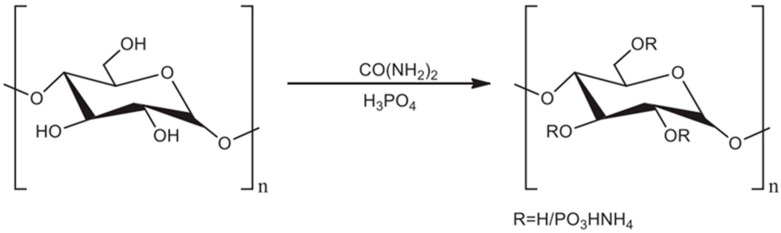
Phosphorylation of polysaccharides.

**Figure 5 polymers-14-04161-f005:**
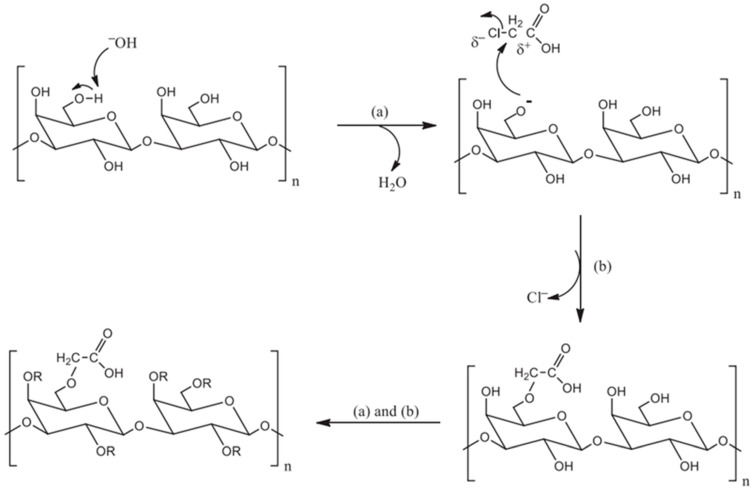
Carboxymethylation of polysaccharides. (**a**): Alkoxide, (**b**): Monochloroacetic acid.

**Figure 6 polymers-14-04161-f006:**
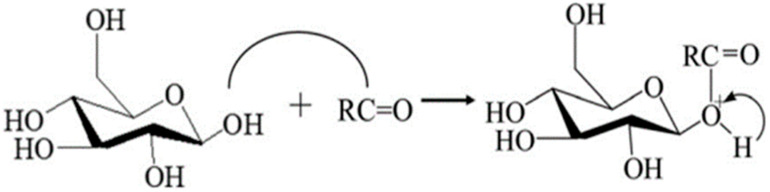
Acetylation of polysaccharides.

**Table 2 polymers-14-04161-t002:** Selenization modification of natural plant polysaccharides (2012–2022).

Source	Extraction Method	B-M_w_ (kda)	A-M_w_ (kda)	Modification Method	Main Modifying Conditions	Content	B-CHO (%)	A-CHO (%)	Biological Activity	Refer
Ulmus pumila L (PPU)	Hot-water extraction	2.697 × 10^9^	3.977–6.528 × 10^9^	Nitric acid/sodium selenite method	PPU (1 g) HNO_3_ (50 mL, 5%) Na_2_SeO_3_ (200, 400, 600, 800, 1000 mg) Reaction at room temperature for 24 h	3.24–13.19 mg/g	88.87	N	Antioxidant	[57]
Artemisia sphaerocephala (PAS)	Hot-water extraction	69.6	2.5–58.6	Glacial acetic acid-selenous acid method	PAS (300 mg) DMSO (30 mL) H_2_SeO_3_ (40 mL) Reaction at (50, 70, 90 °C) for (4, 6) h	139–8744 μg/g	N	N	Immunomodulatory	[59]
Sagittaria sagittifolia L. (PSSP)	Subcritical extraction	47.12	16.82	Nitric acid/sodium selenite method	PSSP (500 mg) HNO_3_ (50 mL, 0.6%) BaCl_2_ (5 mL, 0.1 M) Na_2_SeO_3_ (0.5 g) Reaction at 75 °C for 8 h	2.89 μg/g	77.67	82.26	Antioxidant Immunomodulatory	[60]
Artemisia sphaerocephala (ASP)	Microwave-assisted method	73.5	11.4–331.5	Selenium oxychloride method	ASP (500 mg) Formamide (20 mL) NaHSeO_3_ (1 g) SOCl_2_ (10 mL) Reaction at 60 °C for 10–60 min	264- 22400 μg/g	N	N	Antitumor	[61]
Chinese angelica (CAP)	Hot-water extraction	N	N	Nitric acid/sodium selenite method	CAP (500 mg) HNO_3_ (50 mL, 5%) Na_2_SeO_3_ (200, 300, 400 mg) Reaction temperature (50, 70, 90 °C) Reaction time (6, 8, 10 h)	6.41–12.98 mg/g	92.7	23.5–63.2	Immunomodulatory	[62]
Lycium barbarum (LBP)	Hot-water extraction	N	N	Nitric acid/sodium selenite method	LBP (500 mg) Na_2_SeO_3_ (200, 300, 400 mg) Reaction temperature (50, 70, 90 °C) Reaction time (6, 8, 10 h)	7.65–13.66 mg/g	87.1	19.2–44.6	Antioxidant	[63]
Garlic (GPS)	Hot-water extraction	N	N	Nitric acid/sodium selenite method	GPS (500 mg) Na_2_SeO_3_ (200,300, 400 mg) Reaction temperature (50, 70, 90 °C) Reaction time (6, 8, 10 h)	6.21–12.49 mg/g	94.5	21.2–56.9	Immunomodulatory	[64]
Codonopsis pilosula pectic (CPP1b)	Hot-water extraction	148	195	Nitric acid/sodium selenite method	CPP1b (100 mg) HNO_3_ (20 mL) Ratio of CPP1b to Na_2_SeO_3_ (2:1, 2:1.6, 2:2) Reaction temperature (60, 70, 80 °C) Reaction time (5, 7, 9 h)	94.06–478 μg/g	N	N	Antitumor	[65]
Atractylodes macrocephala (AMP)	Hot-water extraction	N	N	Nitric acid/sodium selenite method	AMP (500 mg) Na_2_SeO_3_ (200,300, 400 mg) Reaction temperature (50, 70, 90 °C) Reaction time (6, 8, 10 h)	6.12–12.23 mg/g	84	36.1–62.99	Immunomodulatory	[66]
Radix hedysari (RHP)	N	N	27.7–62.7	Nitric acid/sodium selenite method	RHP (400 mg) HNO_3_ (40 mL, 0.6%) Reaction at 65°C for (3, 5, 10, 15 h)	1.04–3.29 mg/g	N	N	Antioxidant	[67]
Artemisia sphaerocephala (ASP)	Microwave-assisted extraction	73.48	17.36- 46.67	Nitric acid/sodium selenite method	ASP (500 mg) HNO_3_ (50 mL, 0.8%) H_2_SeO_3_ (1.0 g) BaCl_2_ (1.65 g) Reaction at 60 °C for (15–480) min by 300W ultrasonic powers	111–264 μg/g	90.2	69.8–86.8	Antitumor	[68]
Lily (LP)	Hot-water extraction	N	N	Nitric acid/sodium selenite method	LP (500 mg) HNO_3_ (50 mL, 0.5%) Na_2_SeO_3_ (200,300, 400 mg) Reaction temperature (50, 70, 90 °C) Reaction time (6, 8, 10 h)	11.8–39.2 mg/g	N	56.31–77.14	Immunomodulatory	[69]
Artemisia sphaerocephala (ASP)	Microwave-assisted extraction	73.48	11.41–54.07	Nitric acid/sodium selenite method	ASP (500 mg) HNO_3_ (50 mL, 0.4, 0,7, 1.2%) BaCl_2_ (1.65 g) Na_2_SeO_3_ (1000 mg) Reaction temperature (60, 70, 80 °C) Reaction time (6 h)	168–1703 μg/g	90.2	N	Antitumor	[70]
Glycyrrhiza uralensis (GUP)	Hot-water extraction	38.032	5.8	Nitric acid/sodium selenite method	GUP (500 mg) HNO_3_ (50 mL, 0.5%) Na_2_SeO_3_ (200 mg) Reaction at 70 °C for 8 h	1.34 mg/g	N	N	Antioxidant	[71]
Tea (TPS)	Hot-water extraction (pretreatment degreasing)	N	N	Nitric acid/sodium selenite method	TPS (500 mg) HNO_3_ (50 mL) Na_2_SeO_3_ (0.5 g) BaCl_2_ (0.1 mol/L) Reaction at 75 °C for 8 h	2.12 mg/kg	62.23	60.26	Hypoglycemic activity	[72]
Alfalfa (RAPS)	Hot-water extraction (pretreatment degreasing)	15.8	11.0	Nitric acid/sodium selenite method	RAPS 50 mg HNO_3_ (10 mL,0.6%) Na_2_SeO_3_ (50 mg) Reaction at 70 °C for 10 h	320 μg/g	97.1	N	Antioxidant Antitumor	[73]
Momordica charantia L. (MCPIIa)	Hot-water extraction	13	40.038	Nitric acid/sodium selenite method	MCPIIa (10 mg) Na_2_SeO_3_ solutions (2.5–10 mL, 0.05 M) Ascorbic acid solution (8 mL, 0.1 M). Reaction at 28 °C for 12 h	445.0 μg/g	93.99	92.12	Hypoglycemic activity	[74]
Codonopsis pilosula (CPPS)	Hot-water extraction	345	230.6	Nitric acid/sodium selenite method	The ratio of sodium selenite to CPPS was 0.6:1 Reaction at 70°C for 8 h	11.86 mg/g	98.86	N	Immunomodulatory	[75]
Astragalus (APS)	N	12.314	10.042	Nitric acid/sodium selenite method	APS (500 mg) HNO_3_ (50 mL, 5%) BaCl_2_ (1 g) Na_2_SeO_3_ (400 mg) Reaction at 70 °C for 6 h	1.75 mg/g	N	N	Antioxidant	[76]
Ribes nigrum L. (RCP)	Ultrasonic-assisted extraction	20.4	9.09–12.9	Nitric acid/sodium selenite method	RCP (500 mg) BaCl_2_ (2.5 g) HNO_3_ (250 mL, 0.5 %) Na_2_SeO_3_ (500 mg) Reaction at (50, 80) °C for (3, 5, 7) h by (500, 800) W ultrasonic powers	70–480 μg /g	85.5	82.12–83.58	Hypoglycemic activity	[77]
Dandelion roots (DRP)	Ultrasonic-assisted extraction	8.7	5.6–7.9	Nitric acid/sodium selenite method	DRP (500 mg) HNO_3_ (50 mL, 0.5%) BaCl_2_ (700 mg) Na_2_SeO_3_ (500 mg) Reaction at 40 °C for 4 h Reaction at 60 °C for 8 h	170–710 μg/g	94.24	96.31–96.72	Immunomodulatory Antioxidant	[78]
Yam (YPS)	Hot-water extraction	N	N	Nitric acid/sodium selenite method	YPS (500 mg) HNO_3_ (8 mL, 0.5%) Na_2_SeO_3_ (25–50 mg) Reaction at 75 °C for 8 h	715–1545 mg /kg	N	N	Immunomodulatory	[79]
Purslane (PSPO)	Hot-water extraction	N	N	Nitric acid/sodium selenite method	PSPO (300 mg) HNO_3_ (20 mL, 5%) Na_2_SeO_3_ (45 mg) Reaction at 75 °C for 8 h	753–1325 mg/kg	N	N	Immunomodulatory	[80]
Garlic (GPS)	Hot-water extraction	N	N	Nitric acid/sodium selenite method	GPS (500 mg) HNO_3_ (0.5%,50 mL) Na_2_SeO_3_ (400 mg) Reaction at 70 °C for 6 h	10.5–38.3 mg/kg	95.26	31–54.8	Antioxidant Immunomodulatory	[81]
Rose laevigata Michx fruit (PPRLMF-2)	Hot-water extraction	137.1	82.158	Nitric acid/sodium selenite method	PPRLMF-2 (500 mg) HNO_3_ (0.5%) Na_2_SeO_3_ (400 mg) Reaction at 70 °C for 8 h	862 μg/g	N	94.2	Immunomodulatory	[82]

N: not referred; B-M_w_: M_w_ before modification; A-M_w_: M_w_ after modification; B-CHO: carbohydrate content before modification; A-CHO: carbohydrate content after modification.

**Table 3 polymers-14-04161-t003:** Phosphorylation modification of natural plant polysaccharides (2012–2022).

Source	Extraction Method	B-M_w_ (kda)	A-M_w_ (kda)	Modification Method	Main Modifying Conditions	DS	B-CHO (%)	A-CHO (%)	Biological Activity	Refer
Amana edulis (AEPs)	Acidic extraction Hot-water extraction	N	N	Phosphate method	CP (500 mg) Sodium tripolyphosphate 8.57 g Sodium trimetaphosphate 1.43 g Double-distilled water 100 mL 5% of sodium sulfate Reaction at 90 °C for 4 h	0.42–0.54	56.53–65.61	53.35–63.17	Antioxidant	[23]
Orchis chusua D. Don (SP)	Hot-water extraction	369	355	Phosphate method	SP (200 mg) Sodium tripolyphosphate (1.74 g) Sodium trimetaphosphate (0.286 g) Double-distilled water (20 mL) Sodium sulfate (1 g) Reaction at 90 °C for 2 h	0.32	47.93	63.86	Antioxidant Probiotic ability	[53]
Cyclocarya paliurus (CP)	Hot-water extraction	139	155	Phosphate method	CP (500 mg) Sodium tripolyphosphate (5.58 g) Sodium trimetaphosphate (2.23 g) Double-distilled water (100 mL) 5% of sodium sulfate Reaction at 65 °C for 5 h	0.14	55.13	43.58	Antioxidant	[83]
Dictyophora indusiata (DIP)	Acidic extraction (pretreatment degreasing)	N	65	Phosphoric acid method	DIP (4 g) 200 mL of DMSO containing 8 M urea Phosphoric acid (40 mL) Reaction at 100 °C for 6 h	0.206	N	N	Antioxidant Antitumor	[84]
Pumpkin (N)	Hot-water extraction	10.1	9–17	Phosphorus oxychloride method	Polysaccharides (500 mg) Pyridine (30 mL) N, N-Dimethylformamide (20 mL) POCl_3_ (4 mL) Reaction at 60 °C for 3 h	0.33–0.52	97.42	44.7–65.38	Antioxidant	[85]
Artemisia sphaerocephala (ASP)	Microwave-assisted extraction	73.48	65.85–137.7	Phosphorus oxychloride method	ASP (500 mg) N, N-Dimethylformamide (20 mL) Pyridine (10 mL) Time (1–6 h) Temperature (0–60 °C)	034–0.54	90.2	N	n	[86]
Chrysanthemum indicum (CIPS)	Hot-water extraction	N	N	Phosphate method	Ratio of sodium trimetaphosphate to sodium tripolyphosphate (5:2) Time (8 h) Temperature (70 °C)	0.317	N	N	Antiviral	[87]
Radix Cyathulae officinalis Kuan (RCPS)	Hot-water extraction	N	N	Phosphate method	Ratio of sodium trimetaphosphate to sodium tripolyphosphate (1:2, 1:4, 1:8) Time (2, 4, 8 h) Temperature (65, 80, 95 °C)	0.31–0.77	96.6	78.9–96.6	Antiviral Immunomodulatory	[88,89,90]
Pumpkin (N)	Hot-water extraction	N	N	Phosphorus oxychloride method	Polysaccharides (500 mg) Pyridine (7.5 mL) Chlorosulfonic acid (1 mL) N, N-Dimethylformamide (7.5 mL) Reaction at 80 °C for 3 h	0.01–0.02	81	69–75	Antioxidant	[91]

N: not referred; B-M_w_: M_w_ before modification; A-M_w_: M_w_ after modification; B-CHO: carbohydrate content before modification; A-CHO: carbohydrate content after modification.

**Table 4 polymers-14-04161-t004:** Carboxymethylation modification of natural plant polysaccharides (2012–2022).

Source	Extraction Method	B-M_w_ (kda)	A-M_w_ (kda)	Modification Method	Main Modifying Conditions	DS	B-CHO (%)	A-CHO (%)	Biological Activity	Refer
Amana edulis (AEPs)	Acidic extraction Hot water extraction	N	N	Solvent method	AEPs (500 mg) Isopropanol (12.5 mL) React at 25 °C for 3 h followed by reaction at 60 °C which lasted for 1.5 h	0.605–0.783	56.53–65.61	52.22–60.45	Antioxidant	[23]
Cucumber (N)	Hot-water extraction	N	N	Water media method	Polysaccharide (500 mg) NaOH (6.5 g, 20%) Chloroacetic acid (4 g, 30%) Reaction at 70 °C for 4 h	0.18	20.6	44.07	Antioxidant	[45]
Orchis chusua D. Don (SP)	Hot-water extraction	369	68	Water media method	SP (200 mg) NaOH (15.2 mL, 20%) Chloroacetic acid (6.2 mL, 30%) Reaction at 70 °C for 2 h	0.13	47.93	54.71	Antioxidant	[53]
Cyclocarya paliurus (CP)	Hot-water extraction	139.6	175.4	Solvent method	CP (300 mg) Isopropanol (20 mL) Carboxymethylation reagent (10 mL, 20% NaOH, 3 g chloroacetic acid, and 20 mL isopropanol) Reaction at 55 °C for 5 h	0.29	N	N	Antioxidant Antitumor	[54]
Blackcurrant fruits (RNP)	Ultrasonic-assisted enzymatic extraction	8.093	11.036–12.548	Solvent method	RNP (100 mg) Isopropanol (5 mL) NaOH (2.0 mL, 20%) The resulting solution was heated at different temperatures (60–100 °C) at various ratios of MCA to polysaccharide (8:1–30:1) for various periods (20–60 min)	0.44–1.1	51.95	47.61–52.47	Antioxidant Anti-lipid peroxidation Anti-hemolytic	[93]
Cyclocarya paliurus (CP)	Hot-water extraction	N	1.03–1.08 × 10^3^	Water media method	CP (500 mg) NaOH (38 mL, 20%) Chloroacetic acid (1, 2, 3 g) Reaction at 55 °C for 5 h	0.025– 0.193	60.62	55.13–58.16	Antioxidant	[94]
Bitter gourd (P)	Hot-water extraction	N	N	Water media method	P (1000 mg) NaOH (85 mL, 20%) Chloroacetic acid (15 mL, 4 mol/L) react at 55 °C for 5 h	0.89	74	43.5	Antioxidant	[95]
Garlic (P)	Hot-water extraction (Pretreatment degreasing)	N	N	Water media method	P (800 mg) NaOH (12.5 g, 20%) Chloroacetic acid (7.5 g, 30%) Reaction at 70 °C for 4 h	0.92	76.67	60.27	Antioxidant	[96]
Schisandra (SP)	Hot-water extraction	143	N	Solvent method	SP (500 mg) Isopropanol (20 mL) Chloroacetic acid (0.83 g) Reaction at 62.67 °C for 4.27 h	0.88	82.5	89.5	Immunomodulatory	[97]

N: Not Referred; B-M_w_: M_w_ before modification; A-M_w_: M_w_ after modification; B-CHO: Carbohydrate content before modification;A-CHO: Carbohydrate content after modification.

**Table 5 polymers-14-04161-t005:** Acetylation modification of natural plant polysaccharides (2012–2022).

Source	Extraction Method	B-M_w_ (kda)	A-M_w_ (kda)	Modification Method	Main Modifying Conditions	DS	B-CHO (%)	A-CHO (%)	Biological Activity	Refer
Orchis chusua D. Don (Salep) (SP)	Hot-water extraction	369	331	Acetic anhydride method	SP (200 mg) pH 9.0 Acetic anhydride (3 mL) pH of the reaction was maintained at 8.0–8.5 for 30 min at 60 °C	0.56	58	47.93	Probiotic ability	[53]
Bitter gourd (P)	Hot-water extraction	N	N	Acetic anhydride method	P (500 mg) pH 9.5 Acetic anhydride (0.6 mL) reacted at room temperature for 1 h	0.27	74	62.3	Antioxidant	[95]
Cyclocarya paliurus (CP)	Hot-water extraction	900	1.05 × 10^3^	Acetic anhydride method	CP (500 mg) Acetic anhydride (1 mL) pH 8.0–8.5	0.13	60.62	64.89	Immunomodulatory	[98]
Cyclocarya paliurus (CP)	Hot-water extraction	N	1.05 × 10^3^ 1.08 × 10^3^ 1.09 × 10^3^	Acetic anhydride method	CP (500 mg) pH 9.0 Acetic anhydride (1, 4, 6 mL) pH of the reaction was maintained at 8.0–8.5 for 4 h	0.13–0.57	60.62	64.89–66.91	Antioxidant	[99]
Artemisia sphaerocephala Krasch. (ASKP)	Hot-water extraction	525.9	321.7 446.8 799.0 1329	Acetic anhydride method	ASKP (1000 mg) pH 8.0–8.5 Acetic anhydride (2.5 mL) Reaction at 25 °C for 2 h	0.04–0.42	N	N	Emulsifying	[100]
Cyclocarya paliurus (CCP)	Hot-water extraction (pretreatment degreasing)	38.4	30.7	Acetic anhydride method	CCP (200 mg) pH 8.0–8.5 Acetic anhydride (0.8 mL) Reaction at 40 °C for 2 h	0.18	94.94	90.82	Antioxidant	[101]
Garlic (PS)	Hot-water extraction (pretreatment degreasing)	N	N	Acetic anhydride method	PS (1500 mg) pH 11.0 Acetic anhydride (5 mL) pH of the reaction was maintained at 7–11 for 2.5 h at 30 °C	0.5	N	N	Antioxidant	[102]
Millettia speciosa Champ (MSCP)	Hot-water extraction	15.6	9–18.8	Acetic anhydride method	MSCP (300 mg) pH 8.0 Acetic anhydride (1, 3, 5 mL) pH of the reaction was maintained at 8.0–8.5 for 2 h	0.1–0.56	80.25	70.43–76.21	Emulsifying Antioxidant	[103]
Sugar beet pulp (ASP2)	Acidic extraction	238	336	Acetic anhydride method	ASP2 solution (1.5% *w*/*w*) pH 8.0 Acetic anhydride (1–6% *w*/*w*) pH of the reaction was maintained at 7.0–7.5 for 30 min	0.86	N	N	Emulsifying	[104]

N: Not Referred; B-M_w_: M_w_ before modification; A-M_w_: M_w_ after modification; B-CHO: Carbohydrate content before modification; A-CHO: Carbohydrate content after modification.

## Data Availability

The datasets generated for this study are available on request to the corresponding author.

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
