# Peer review of "Chemical Modification, Characterization, and Activity Changes of Land Plant Polysaccharides: A Review"

_polymers, 2022, doi:10.3390/polym14194161_

Round 1

Reviewer 1 Report

Being a review paper, the manuscript needs some improvements. Authors should not mention the research progress of chemical modification of plant polysaccharides during the period 2012-2022 because the paper was submitted in 2022, not 2023, and authors should do more literature review until the end of 2022 if they still insist on the statement.

Authors should review the chemical modification of plant polysaccharides using a solventless method because it is an environmentally friendly method and not harmful to humans.

Authors should justify why the review excludes marine plant polysaccharides and fungi plant polysaccharides because marine and fungi plants are also important sources for polysaccharides; otherwise, authors must include marine plant and fungi plant polysaccharides in the manuscript.

Types of biomasses and their impact on the extraction of polysaccharides, as well as possible pre-treatments, should also be included in the manuscript.

Authors should review the related applications of modified polysaccharides, not only mentioned for food, nutraceuticals, and pharmaceuticals.

Properties of the modified polysaccharides obtained, and the effect of each method used in these properties should be addressed.

A comprehensive discussion of the advantages of selected modification methods over other methods to modify polysaccharides and future perspectives should be included.

Author Response

Dear Reviewer, According to your request, I have responded to your questions one by one.

Reviewer 2 Report

The article on the topic "Chemical modification, characterization and activity changes of plant polysaccharides: a review" is a qualitatively done review of the modification of plant polysaccharides isolated by almost one method, specifically water extraction followed by ethanol precipitation. The authors focus on five types of modification, the main purpose of which is to obtain stronger biologically active substances from plant polysaccharides. The review covers the last 10 years. The strength of this article is the authors' attempt to analyze and present in an attractive way the idea of ​​modification methods, changing the properties of modified polysaccharides and their purpose in the food and pharmaceutical industries. A good Figure 1 would be a great graphic annotation for this article.

But there are recommendations for improving the article:

1. It is necessary to include in the text of the article information (possibly in a short separate section) on the classification of the considered polysaccharides, indicate the range of their content, despite the variety of plant materials. This kind of information will emphasize the significance of the polysaccharide class and will indicate the individuality of the presented review.

2. It is necessary to add to the text information about the methods (briefly, but it can be done in a short separate section) for the isolation of polysaccharides, indicate the range of yields during isolation by extraction with water followed by precipitation with ethanol.

3. The abstract lacks information that five types of modifications of polysaccharides isolated from plant materials by water extraction followed by ethanol precipitation will be considered.

4. I recommend that, at least in the introduction, give the number of types of plant materials considered in this article. Undoubtedly, such a variety will attract readers.

5. As a reviewer, I accept any way to supplement the submitted review.

Author Response

(The authors gave the same response as above.)

Round 2

Reviewer 1 Report

This manuscript has been carefully revised according to reviewers' comments; thus, it is recommended for acceptance at the current stage, but the English grammar and sentence structure of the manuscript needs to be revised and corrected.

Author Response

The grammar of the paper was revised according to the reviewer's request.
